# Comparative Investigation of Cutting Devices on Bone Blocks: An SEM Morphological Analysis

**Roberto Lo Giudice** [1,*] **, Francesco Puleio** [2] **, David Rizzo** [2] **, Angela Alibrandi** [3] **,
Giorgio Lo Giudice** [4] **, Antonio Centofanti** [2] **, Luca Fiorillo** [2] **, Debora Di Mauro** [2] **and
Fabiana Nicita** [2]

1   Department of Clinical and Experimental Medicine, Messina University, 98100 Messina, Italy
2   Department of Biomedical and Dental Sciences and Morphofunctional Imaging, Messina University, 98100
    Messina, Italy; francesco.puleio@live.it (F.P.); dr.davidrizzo@libero.it (D.R.);
    centofantiantonio@gmail.com (A.C.); lucafiorillo@live.it (L.F.); debora.dimauro@unime.it (D.D.M.);
    fabin92@hotmail.it (F.N.)
3   Department of Economics, Section of Statistical and Mathematical Sciences, Messina University,
    98100 Messina, Italy; angela.alibrandi@unime.it
4   Private medicinae doctor; 98100 Messina, Italy; giorgio.logiudice@gmail.com
*   Correspondence: rlogiudice@unime.it; Tel.: +39-3934-39997

**Abstract: Background**: Bone regeneration is a reliable technique when the bone volume is insufficient to provide a functional and aesthetic outcome in surgery and implantoprosthesis procedures. When bone blocks are used but do not match the shape of the defect, the block must be adapted. The aim of our research was to evaluate, by Scanning Electron Microscopy (SEM) morphological observation, how different cutting devices modify the bone surface. **Method**: Four equine bone blocks were divided into 15 cubic shape samples with ultrasonic and sonic tips, as well as diamond, tungsten carbide, and Lindemann burs. The uncut surface of the obtained bone block was used as a control. Two observers independently analyzed the SEM observation recording, including cut precision, depth of incision, thermal damages, and presence of bone debris. For each group, sharpness, depth, carbonization, and bone debris were expressed as mean values. **Results**: The osteotomy performed with an ultrasonic tip shows the best results, preserving the bone morphology in both quantitative and qualitative analyses. The bone surface appeared sufficiently clean from debris and showed a reduced presence of carbonization. **Conclusion**: The shaping of the bone block as in vivo osteotomy respects the bone morphology and allows it to achieve the relevant biological and clinical outcome.

**Keywords:** cutting devices; bur; piezosurgery; sonosurgery; bone block

## 1. Introduction

Bone substitute materials can be categorized into four groups depending on their origin: autologous from the same subject, allogenic from another individual within the same species, xenogeneic from another species, and alloplastic that are synthetically produced [1].

Autologous bone contains both active bone cells and growth factors that can promote bone regeneration [2], but its use in clinical practice is limited due to post-operative morbidity from graft infections and/or neurologic risks related to bone harvesting in intra-oral sites [3].

Allografts and xenografts are osteoconductive but not osteoinductive [4,5], unlike other xenograft equine-derived biomaterials. They are used successfully in several fields of oral surgery [6–14] and should be preferred for being resistant to prion infections (i.e., Bovine Spongiform Encephalopathies) [15,16].

Moreover, nonantigenic equine bone retains type I collagen, along with a cellular matrix that contains a high amount of growth factors such IGF II, TGF-beta, IGF I, PDGF, bFGF, and BMPs [17]. Many vitro studies show the positive effects of collagen in the production of new bone [7,18–20].

Equine bone substitute blocks may represent an alternative to particulate grafting materials for many types of augmentation procedures, since it ensures a source of osteoinductive growth factors and a rigid structure for mechanical support [21–24].

In other studies, the resorption rates reported confirmed the excellent volumetric stability and great flexibility of autologous bone blocks in all clinical situations, despite intra- and post-operative complications [25–28].

Particulate xenogeneic bone substitutes in combination with collagen membranes are indicated for deficient ridge contour augmentation but, due to their mechanical instability, need a rigid membrane to obtain three-dimensional augmentation [29,30].

The use of xenogeneic bone blocks allows clinicians to overcome the poor three-dimensional stability of particulate substitutes without the necessity of a rigid membrane, providing good long-term results [2,31].

The bone blocks provided in cubic or parallelepiped shape do not usually match the three-dimensionality of the bone defects, resulting in the necessity to model the blocks [32].

The shaping can be performed before or during the surgery. Should the clinician provide a Cone Beam Computed Tomography (CBCT) prior to surgery, the industry can send a preshaped and sterilized bone graft. Alternatively, the clinician can shape the bone block intra- or peri-operatively by hand, or by using a numerical control milling machine (CNC).

Hand shaping could be performed with rotating instruments or vibrating instruments.

Rotating instruments, mounted on high-speed or low-speed devices, use burs for different materials and shapes to perform an osteotomy, and vibrating instruments are divided into piezoelectric instruments and sonic instruments.

Piezoelectric instruments exploit the dimensional deformation of piezoceramic disks subjected to the switch of the electric charge to make a tip vibrate. The tips could be different in shape and material, with a precise indication for each.

An osteotomy is performed with bone micronization that is produced by mechanical shockwaves, with a linear vibration ranging from 24 to 36 kHz and an amplitude varying from 20 to 200 mm. The main features of this device are represented by a micrometric cut, namely the selective activity on the mineralized tissues and the positive influence of the ultrasonic cut on bone healing [33].

The air-driven sonic osteotome performs an osteotomy with a vibration ranging from 3 to 6 kHz and an amplitude varying from 200 to 300 mm. Sonic tips rotate with a circular tapping motion and are oriented by the friction into the osteotomy line [34].

The cutting action that causes bone trauma may produce carbonization and debris, which may interfere with the healing response in all surgical procedures. In vivo osteotomies should respect bone morphology to achieve the best biological and clinical outcome [35].

The bone substitute is placed in direct contact with the native bone that should be regenerated, so the ideal interface between the two should mimic the natural bone, allowing cell colonization instead of a resorptive process that will slow down the healing process [36].

The knowledge of the advantages and limitations of each device and technique could provide a better clinical outcome.

The aim of our SEM evaluation is to observe the bur/tip-bone contact area and to evaluate the macro- and microstructural changes in the bone structure due to surgical trauma on cancellous bone.

## 2. Results

### 2.1. Quantitative Analysis

For each group, observational parameters (sharpness, depth, carbonization, and bone debris) were recorded. The data (mean value and standard deviation) are summarized in Table 1.

**Table 1.** Per group observational parameters (mean values and standard deviation).

| Burs/Tips | Groups | Observational Parameters | | | |
|---|---|---|---|---|---|
| | | Sharpness (0–3) | Depth (0–3) | Carbonization (0–3) | Bone Debris (0–3) |
| Stainless steel diamond | A | $1.38 \pm 0.8$ | $0.79 \pm 0.4$ | $0.33 \pm 0.4$ | $1.79 \pm 0.8$ |
| Round tungsten carbide | B | $0.10 \pm 0.3$ | $2.50 \pm 0.8$ | $1.40 \pm 0.5$ | $3.00 \pm 0$ |
| Lindemann stainless steel | C | $0.84 \pm 0.5$ | $1.96 \pm 0.4$ | $0.62 \pm 0.6$ | $2.15 \pm 0.5$ |
| Piezoelectric OT12S | D | $1.63 \pm 0.6$ | $1.00 \pm 0.9$ | $0.38 \pm 0.5$ | $1.31 \pm 0.6$ |
| Sonic SFS 101 | E | $1.38 \pm 0.5$ | $1.50 \pm 0.5$ | $1.00 \pm 0.6$ | $1.56 \pm 0.5$ |
| Control | F | $0.58 \pm 0.8$ | $0.58 \pm 0.5$ | $0.17 \pm 0.4$ | $0.75 \pm 0.4$ |

The Kruskal–Wallis test results were highly significant ($p < 0.05$) for all observational parameters recorded within all groups.

The Mann–Whitney U test with the Bonferroni correction showed a new "adjusted" significance level equal to $0.050/15 = 0.003$.

*2.2. Qualitative Analysis*

2.2.1. Stainless Steel Diamond Bur

The diamond bur group showed a moderate similarity to the original bone morphology. The cut surface showed a moderately irregular cut surface with low cut depth, much debris, and low signs of carbonization. Signs of lacerations of the trabecular structure were evident (Figure 1).

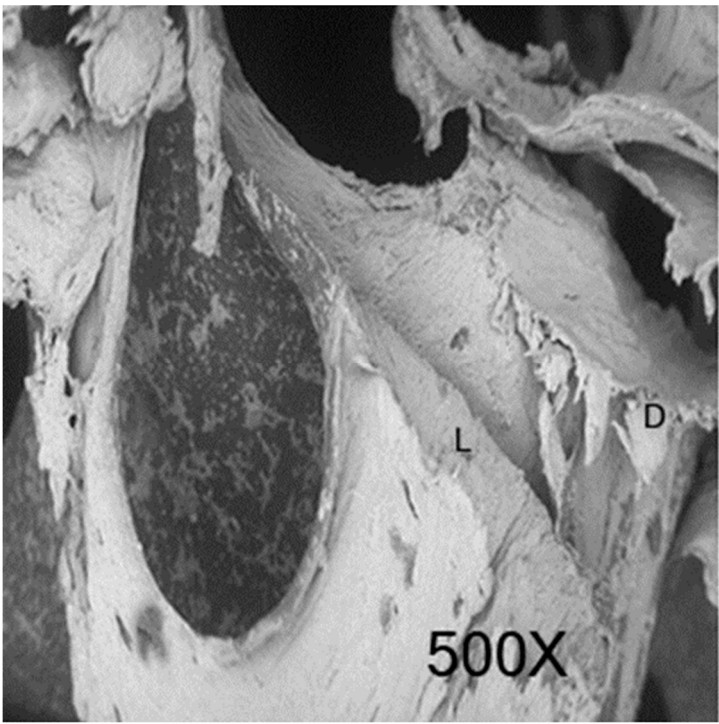

**Figure 1.** Stainless steel diamond bur sample surface (D—debris, L—laceration).

2.2.2. Round Tungsten Carbide Bur

The round tungsten carbide bur showed a subverted bone morphology. The cut surface showed very low sharpness and a high cut depth. A large amount of debris was present but there were low signs of carbonization (Figure 2).

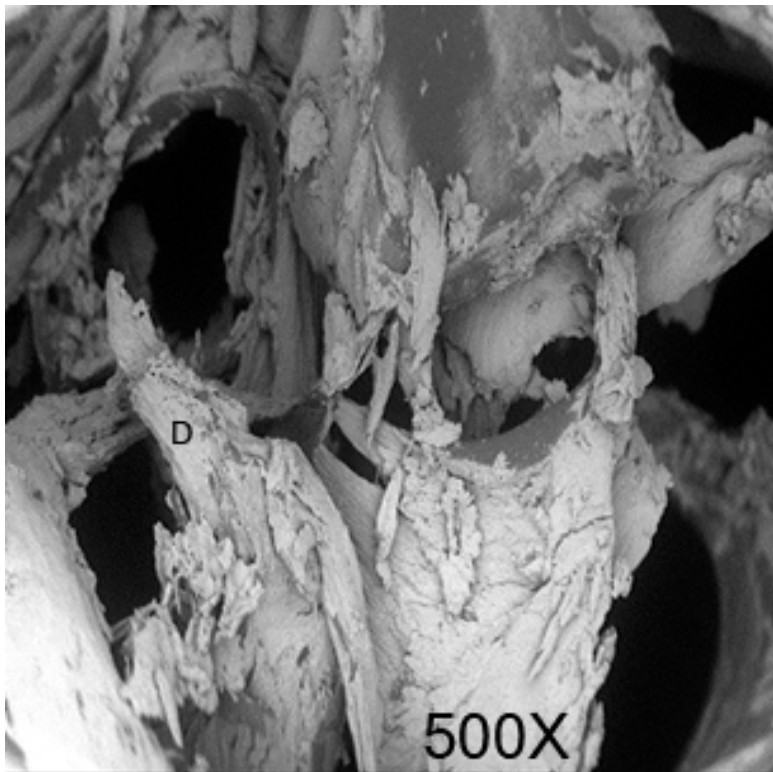

**Figure 2.** Round tungsten carbide bur sample surface (D—debris).

### 2.2.3. Stainless Steel Lindemann Type Bur

The Lindemann type bur showed a cut surface with low correlation to the original morphology. The cut area appeared irregular in depth and shape, with much debris and many signs of carbonization. Microcracks and the exfoliation of bone layers were also visible (Figure 3).

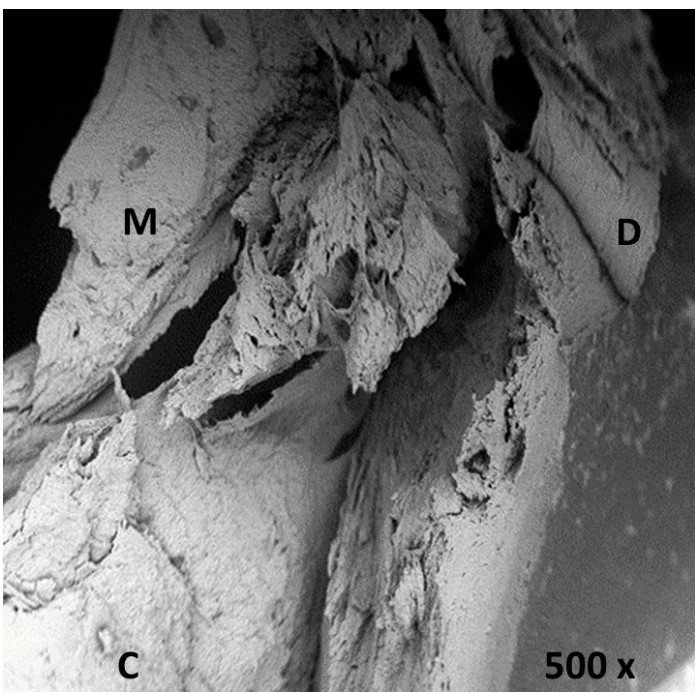

**Figure 3.** Stainless steel Lindemann type bur sample surface (D—debris, C—carbonization, M—microcracks and exfoliation).

### 2.2.4. Piezosurgery OT12S Tip

The Piezosurgery OT12S group showed a cut surface with a high similarity to the original bone morphology. The cut area displayed a low presence of debris, a low cut depth, a moderately irregular shape, and a low presence of bone carbonization (Figure 4).

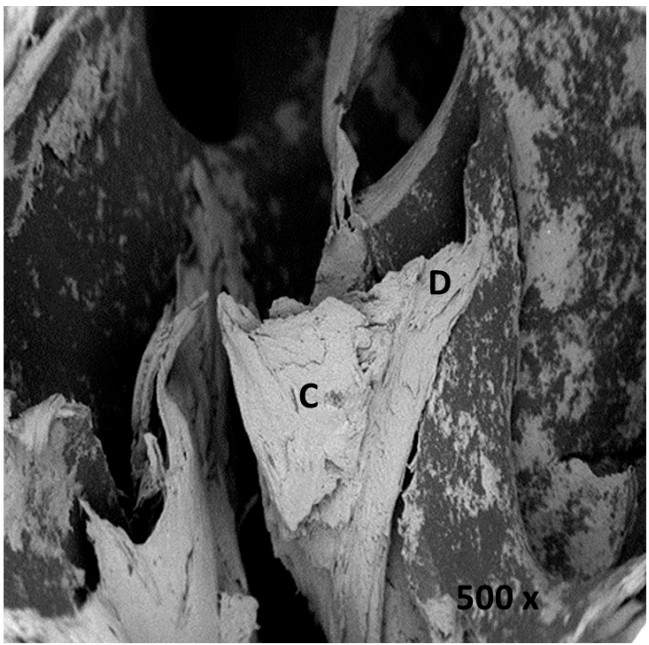

**Figure 4.** Piezosurgery OT12S tip sample surface (D—debris, C—carbonization).

### 2.2.5. Sonosurgery SFS 101

The Sonosurgery SFS 101 group showed a surface with moderate similarity to the original anatomy. The cut surface showed a moderate cut precision and depth, with debris and many signs of carbonization. The bone surface appeared to be smooth and regular with few signs of exfoliation of bone layers (Figure 5).

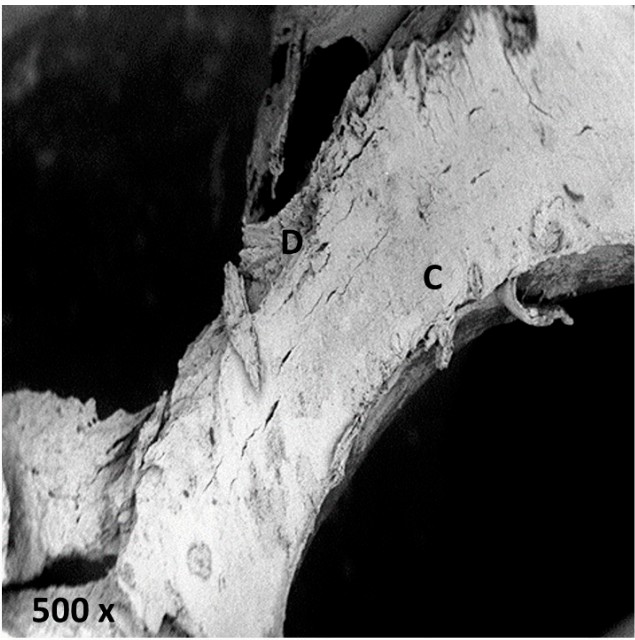

**Figure 5.** Sonosurgery SFS 101 tip sample surface (D—debris, C—carbonization).

### 2.2.6. Control Sample

The control sample group showed a well-represented bone structure with cavities of spherical or ovoid shape, delimited by a dense trabeculation. On the surface it was possible to see collapsed collagen fibers and limited debris (Figure 6).

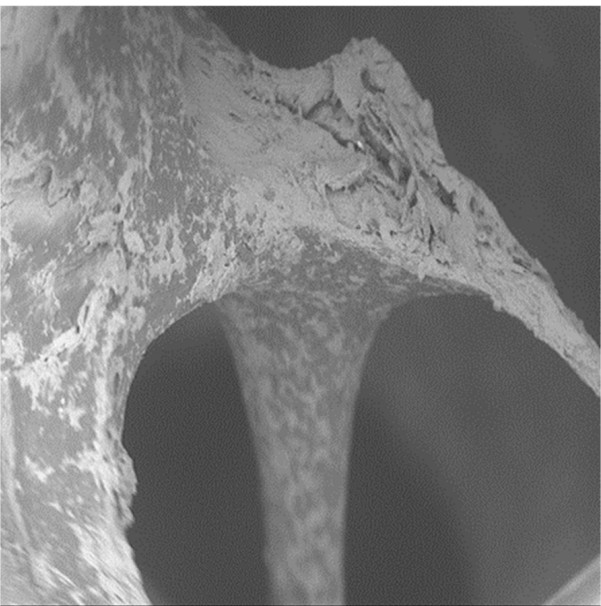

**Figure 6.** Control group sample surface.

## 3. Discussion

When a bone defect is present, it could be necessary to perform a bone augmentation technique to guarantee long-term success and a satisfactory esthetic outcome.

The bone augmentation technique could be performed using a bone substitute, in particulate or in block. To overcome the poor three-dimensional stability of particulate bone, it is suggested to use a bone block [23].

The main advantage of the three-dimensional stability of a bone block is related to the possibility of avoiding the use of a rigid membrane or titanium mesh, especially for the vertical augmentation technique. The main disadvantage of the solidity of a bone block is related to the necessity to match the bone shape to the defect shape [22].

The shaping could be performed by the industry or by hand. In the first case the industry could use a CBCT, provided by the clinician, to perform a preshaped block.

Hand shaping could be performed pre- or intra-operatorial using various instruments. Different instruments will have different effects on the bone morphology, being more or less respectful of the bone anatomy.

When the shaping is performed, the best results have been shown when the presence of collagen and the cellular structure are preserved [37].

Bone anatomical morphology is directly linked to the cellular behavior that resorbs/absorbs and colonizes the bone scaffold [38,39].

Moreover, it has been demonstrated that the presence of debris and thermal damage may induce and increase the inflammatory reaction, that may lead to an increased bone resorption up to a graft failure.

All the cutting devices used in the present research were used under irrigation, mimicking a real clinical situation. Irrigation reduced the heat causing osteonecrosis by lowering the cut surface temperature [40].

The SEM analysis found two types of debris—debris still attached to the bone surface and detached debris forming a smear layer that completely or partially covered the bone surface [41].

The spongious bone used in this experiment mimicked human bone morphology, with numerous large spaces that gave a honeycombed or spongy appearance. The bone matrix, or framework, is organized into a three-dimensional latticework of bony processes, called trabeculae, arranged along lines of stress. The spaces between are often filled with marrow and blood vessels.

The spongious bone of the samples also showed microfractures and exfoliations, while microfractures, sometimes incomplete, were seen in the trabeculae.

The bone block model used is also comparable to human bone morphology, so the effect of the bur/tip could be comparable, and the resulting cellular response could be linked and predicted with the best cut surface that is similar to the morphology of the uncut bone [42].

When evaluating the best performance of the cutting tip, parameters such as high cut sharpness, which allows following a precise line on the osteotomy, and low depth of incision, which represents the three-dimensionality of the damage inducted in bone near specific areas, should be taken into consideration.

The presence of debris should be also minimized to ensure that a resorptive process is not induced by the osteoclastic cellular line, along with signs of carbonization which may induce an increased inflammatory response [33,43–46].

Our in vitro SEM analysis showed that the best result was achieved using a piezosurgery tip. The three-dimensional trabecular and lacunar anatomy was better respected when compared to the control group. The cut surface showed a clean cut with fewer signs of lacerations and a lower presence of bone debris, along with fewer signs of carbonization.

From the comparison between the group 4 samples and the control group samples, it was evident that the cut area showed a low presence of debris, a moderately irregular shape (which was evident in all samples but was linked to the shaping action itself), and a low presence of bone carbonization, which could be linked to the action of the irrigating solution and the piezoelectric movement.

When comparing the shape of the trabecula, the gold standard should be compared to an intact trabecula, with its arc shape preserved and straight with no sign of carbonization or cut line, as evident in the control group samples.

The limited presence of debris may be linked to the cavitation effect of the piezoelectric movement that seemed to remove the debris present more effectively. The sonosurgery results were promising but the depth of incision and the high presence of carbonization may limit its use. Another main limitation was represented by the cooling solution and its sterility. The sonic device uses non-sterile drinking water which, being a hypotonic non-sterile solution, does not favor cell homeostasis, and thus may produce contamination of the surgical area [43].

The best result in the rotating devices group was found when using the stainless steel diamond coated bur, which showed results not comparable to the vibrating group. The worst result was shown when using the low-speed round tungsten carbide bur that subverted the anatomical morphology; this is not suggested by the present study.

The statistical analysis confirmed the high statistical significance ($p < 0.05$) for all the observational parameters recorded within all groups. The Mann–Whitney U test with Bonferroni correction confirmed the relevance of data.

## 4. Materials and Methods

Four equine spongious bone blocks ($10 \times 20 \times 5$ mm) with collagen (OX Block; Osteoxenon; Bioteck; Italy) were divided into 15 samples with a cubic shape ($5 \times 4 \times 5$ mm). To obtain the cubic shape, four osteotomies per sample were performed. A single cutting device was used for each sample. The devices for bone modeling used in this study can be seen in Figure 7.

- Group A: One stainless steel diamond bur truncated conical shape (ISO diameter 016) Komet (Komet, Lemgo, Germany);

- Group B: One round tungsten carbide bur Komet (ISO diameter 018) (Komet, Lemgo, Germany);
- Group C: One stainless steel Lindemann type bur Komet (ISO diameter 016) (Komet, Lemgo, Germany)
- Group D: One Mectron OT12S tip mounted on a Piezosurgery Touch (Mectron, Vicenza, Italy);
- Group E: One Sonosurgery SFS 101 mounted on a Sonosurgery (Komet Dental, Lemgo, Germany).

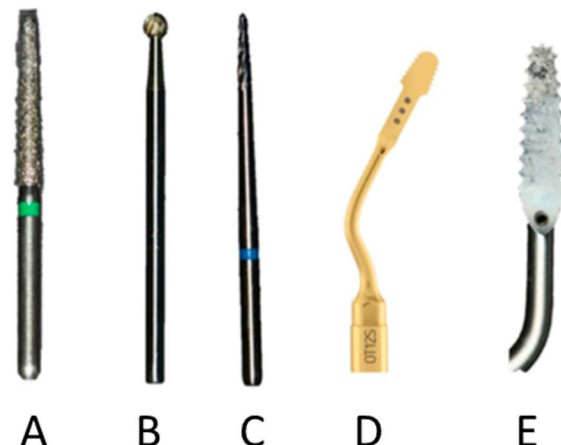

**Figure 7.** Tips/burs used for osteotomy in groups A, B, C, D, E.

The piezosurgery and sonosurgery tips were used with gentle scratching movements and very little pressure (Table 2).

**Table 2.** Features of tips as reported by manufacturers.

| Device | Frequency (kHz) | Insert Tip | Thickness (mm) | Oscillation Amplitude (mm) |
|---|---|---|---|---|
| Piezosurgery Touch | 24 < x < 36 | OT12S | 0.25 | 20 < x < 60 |
| Sonosurgery | 5 < x < 6.5 | SFS101 | 0.25 | 240 |

The Diamond bur and Lindemann bur were mounted on a high-speed handpiece (400,000 rpm) (Bien Air, Dental SA; Bienne, Switzerland).

The round tungstein carbide was mounted on a low-speed handpiece (20,000–40,000 rpm) (W&H Dentalwerk Burmoos GmbH, Bad Hofgastein, Austria).

The uncut upper and lower part of the bone block was used as a control (Group F).

Four SEM observations were performed for each sample in the test area as well as two observations in the control area.

One operator, who was an oral surgeon with more than 10 years of routine practice in oral surgery and the usage of ultrasonic devices, was recruited for the sample creation.

The block perimeter was previously marked with a pencil on the surface by using a titanium template.

All osteotomies were performed following the manufacturer's instructions, and conducted under irrigation with cooled 0.9% sodium chloride solution.

Each device and bur/tip was used in accordance with the manufacturer's recommended settings in order to obtain cuts on bone blocks with the following characteristics:

- Minimal percentage variance between osteotomic track thickness and tip thickness;
- Osteotomized bone surface as smooth as possible;
- Microarchitecture integrity, limiting the presence of bone debris and avoiding thermal injuries to the bone; and
- Cutting time kept as short as possible.

Samples were divided in groups of three for each bur/tip used.

The 15 samples underwent SEM (Phenom Pro 5; Phenom-World B.V.; Eindhoven; Netherlands) observation to directly observe the macroscopical and microscopical structure.

Specimen surfaces were analyzed at a 3072 × 2304 pixel resolution with a magnification of 500× ± 50× [36].

Within the same area of the sample, observations were conducted by taking random references from three default areas that were 500 μm$^2$.

Two calibrated observers independently analyzed the images on a high-definition 24-inch screen. The cut surface of each sample was randomly observed in three areas.

The observational parameters were recorded following the scheme proposed by Romeo et al. as follows:

- Cut precision (sharpness);
- Depth of incision (depth);
- Peripheral thermal damages (carbonization); and
- Presence of bone debris (bone fragments).

For each parameter a score of 0 (very low grade), 1 (low grade), 2 (moderate grade), or 3 (high grade) was given [39]. Numerical data were expressed as means with standard deviation (S.D.) and the categorical variable "group" was expressed as a number and a percentage.

Each parameter score was numerically evaluated following the scheme proposed in Table 3.

**Table 3.** Characteristics of each value in relation to each observational parameter.

| Observational Parameters | Score 0 | Score 1 | Score 2 | Score 3 |
|---|---|---|---|---|
| Sharpness | Impossible to identify anatomical structure | Low preservation of anatomical structure | Moderate preservation of anatomical structure | High preservation of anatomical structure |
| Depth | Very low incision depth | Low incision depth | Moderate incision depth | High incision depth |
| Carbonization | Absence of carbonization | Low presence of carbonization areas | Moderate presence of carbonization areas | High presence of carbonization areas |
| Bone Fragments | Absence of bone fragments | Low presence of bone fragments | Moderate presence of fragments | High presence of bone fragments |

For sharpness, depth, carbonization, and bone debris, we compared all six groups using the Kruskal–Wallis test. Since the results were highly significant, we performed pairwise comparisons between groups using the Mann–Whitney U test. For these multiple comparisons we had to apply the Bonferroni correction, for which the significance alpha level 0.050 had to be divided by the number of the possible pairwise comparisons (15).

Statistical analyses were performed using the Statistical Package for Social Science, (SPSS) V.17.0, IBM Italy, 20090 Segrate (MI), Italy for Windows package. $p < 0.05$ for two sides was considered to be statistically significant.

## 5. Conclusions

The osteotomy performed with the piezosurgery OT12S tip showed the best results, as it preserved the bone morphology and decreased the presence of microfractures. The bone surface appeared sufficiently clean from debris and with the least presence of carbonization among all groups. The shaping of the bone block in vivo via osteotomy should be performed with the aim of respecting the original bone morphology to achieve the best biological and thus clinical outcome.

**Author Contributions:** Conceptualization, R.L.G. and D.R.; Data Curation, G.L.G.; Validation, A.A.; Formal Analysis: A.C. and D.D.M.; Investigation, F.P., F.N., and L.F.

**Funding:** This research received no external funding.

**Conflicts of Interest:** The authors declare no conflict of interest. The funders had no role in the design of the study; in the collection; analyses; or interpretation of data; in the writing of the manuscript; and in the decision to publish the results.

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
