# Peer review of "Comparative Investigation of Cutting Devices on Bone Blocks: An SEM Morphological Analysis"

_applsci, doi:10.3390/app9020351_

Round 1
Reviewer 1 Report
What type of equine bone has been investigated? long Bone, short, cortical, trabecular, spongious, dense?
The discussion is focused on the need to improve cuts in oral surgery. The bone studied is similar to that found in the human jaw?
The authors use as control a commercial block, 10x20x5, obviously cut from a horse. Therefore, it cannot be an uncut control as the authors indicate. It is not very clear to me that control is the most appropriate, since the commercial bone has had to extract the animal's block in some way.
Put the full name the first time that you indicate an acronym and then always use the acronym. Review the text For example line 15, 54 ...
Change S.E.M. by SEM
Table change; by . Wrong mistake?
Line 97 "......" moderate what? With respect to..?
Figure captions are not the devices, tools, bur or tips ... Change by SEM image of the equine bone cutter by ... .whatever.
Indicate the magnification bar inside the SEM figures.
How do you prepare the samples for SEM? Are covered with Au, Pd, C ...? Are they inmmersed in a resin? ...
Indicate by symbols in the SEM images that corresponds to debris, carbonizations, lacerations .....etc.
Indicate in figure caption 7 that device corresponds to group A, B, C, D, E
Line 165-166. Indicate where the cancellous bone is and what is the trabecular bone. Give a description of the SEM figures more in depth, indicating with symbols the different parts and damages produced when cutting them.
Line 176 is asked to make a deeper comparison / discussion between figure 4 and 6 to see that this is the best device to cut bone. Similarities and dissimilarities.
Line 215 ".." 6 control? Re-write. The authors want to say 6 replicas of cuts for each device ?. Or what is referred to 6. Later on line 230 the authors indicate 10 samples went to SEM, which means 2 samples per device (2x5 = 10) and control samples are not included ?. The number of samples indicated in the different parts of the text do not match. In M & M indicate the authors have 4 blocks (10x20x5) and they are divided into 10 (5x5x5). Or do they mean that each block is divided into 10? It is not very clear how they do it
Author Response
· What type of equine bone has been investigated? long Bone, short, cortical, trabecular, spongious, dense?
1. The bone blocks used is made by spongious bone. This detail is added into text
· The discussion is focused on the need to improve cuts in oral surgery. The bone studied is similar to that found in the human jaw?
2. For the trabecular structure of the bone used, it could be compared to the spongious human bone.This aspect is underlined in the discussion paragraph.
· The authors use as control a commercial block, 10x20x5, obviously cut from a horse. Therefore, it cannot be an uncut control as the authors indicate. It is not very clear to me that control is the most appropriate, since the commercial bone has had to extract the animal's block in some way.
3. Considering that, clinicians use bone blocks from the market, and so assuming this kind of blocks as the standard available, the aim of research was to compare if, among what is commercially available, there were any differences between cutting techniques.
· Put the full name the first time that you indicate an acronym and then always use the acronym. Review the text For example line 15, 54 ...
4. Text revised as requested
· Change S.E.M. by SEM
5. Text revised as requested
· Table change; by . Wrong mistake?
6. Table is corrected
· Line 97 "......" moderate what? With respect to..?
7. The scheme proposed by Romeo et al. was used to evaluate the observational parameters. The score is better described in text.
· Figure captions are not the devices, tools, bur or tips ... Change by SEM image of the equine bone cutter by ... .whatever.
8. Captions were corrected.
· Indicate the magnification bar inside the SEM figures.
9. Magnification inserted as requested
· How do you prepare the samples for SEM? Are covered with Au, Pd, C ...? Are they inmmersed in a resin? ...
10. This kind of SEM does not requires any type of specimen preparation.
· Indicate by symbols in the SEM images that corresponds to debris, carbonizations, lacerations .....etc.
11. Debris, carbonization, laceration, microcracks and exfoliation are now indicated into pictures when mostly evident.
· Indicate in figure caption 7 that device corresponds to group A, B, C, D, E
12. Caption corrected as requested
· Line 165-166. Indicate where the cancellous bone is and what is the trabecular bone. Give a description of the SEM figures more in depth, indicating with symbols the different parts and damages produced when cutting them.
13. Letters were embedded in picture to better let the reader see each effect. A description of the spongious bone is added.
· Line 176 is asked to make a deeper comparison / discussion between figure 4 and 6 to see that this is the best device to cut bone. Similarities and dissimilarities.
14. The paragraph is expanded as requested
· Line 215 ".." 6 control? Re-write. The authors want to say 6 replicas of cuts for each device ?. Or what is referred to 6. Later on line 230 the authors indicate 10 samples went to SEM, which means 2 samples per device (2x5 = 10) and control samples are not included ?. The number of samples indicated in the different parts of the text do not match. In M & M indicate the authors have 4 blocks (10x20x5) and they are divided into 10 (5x5x5). Or do they mean that each block is divided into 10? It is not very clear how they do it
15. The paragraph is rewritten to be more clear.
From the 4 blocks we obtained 15 samples doing 4 incision per side. The upper and lower part of the same samples are uncut and so used as control. For each sample we SEM observed the four cut areas and the two control (uncut upper and lower part) areas.
The reply is also enclosed as word file .

Reviewer 2 Report
The article is well focused and could be published if some changes were made: - The text abuses the short sentences and the paragraphs are very short (one or two lines). The author must correct this aspect. - In the Material and method section, a table must be included with the specific characteristics of each of the values handled for each variable.Author Response
The text abuses the short sentences and the paragraphs are very short (one or two lines).
The text is revised as requested
In the Material and method section, a table must be included with the specific characteristics of each of the values handled for each variable.
A table is included in Material and method section
Round 2
Reviewer 1 Report
I have carefully read the revised paper and believe that it is somewhat better than the previous submission. The critiques addressed by the reviewer were answered for the most part satisfactorily. The authors of the paper have clearly done a lot of work, and the results obtained by them are quite interesting. It reads better now.
Reviewer 2 Report
Now, the paper is ok.